# Disentangled Style Domain for Implicit $z$-Watermark Towards Copyright Protection

**Junqiang Huang**
Department of Computer Science
Fudan University
23210240188@m.fudan.edu.cn

**Zhaojun Guo**
Department of Computer Science
Fudan University
22110240087@m.fudan.edu.cn

**Ge Luo**
Department of Computer Science
Fudan University
gluo18@fudan.edu.cn

**Zhenxing Qian**\*
Department of Computer Science
Fudan University
zxqian@fudan.edu.cn

**Sheng Li**
Department of Computer Science
Fudan University
lisheng@fudan.edu.cn

**Xinpeng Zhang**\*
Department of Computer Science
Fudan University
zhangxinpeng@fudan.edu.cn

## Abstract

Text-to-image models have shown surprising performance in high-quality image generation, while also raising intensified concerns about the unauthorized usage of personal dataset in training and personalized fine-tuning. Recent approaches, embedding watermarks, introducing perturbations, and inserting backdoors into datasets, rely on adding minor information vulnerable to adversarial training, limiting their ability to detect unauthorized data usage. In this paper, we introduce a novel implicit Zero-Watermarking scheme that first utilizes the disentangled style domain to detect unauthorized dataset usage in text-to-image models. Specifically, our approach generates the watermark from the disentangled style domain, enabling self-generalization and mutual exclusivity within the style domain anchored by protected units. The domain achieves the maximum concealed offset of probability distribution through both the injection of identifier $z$ and dynamic contrastive learning, facilitating the structured delineation of dataset copyright boundaries for multiple sources of styles and contents. Additionally, we introduce the concept of watermark distribution to establish a verification mechanism for copyright ownership of hybrid or partial infringements, addressing deficiencies in the traditional mechanism of dataset copyright ownership for AI mimicry. Notably, our method achieves one-sample verification for copyright ownership in AI mimic generations. The code is available at: https://github.com/Hlufies/ZWatermarking

## 1 Introduction

Recent advancements in text-to-image generation technologies [1, 2, 3] have revolutionized art creation by enabling users to replicate the unique styles of artists and art images through simple prompts. Simultaneously, text-to-image personalization technologies [4, 5, 6, 7] make it easy to fine-tune generative models with minimal online personal portfolios, which may not be authorized.

---

\*Corresponding author.

38th Conference on Neural Information Processing Systems (NeurIPS 2024).

However, a question arises: are individual styles and contents entitled to copyright protection? Recent studies [8, 9, 10, 11] indicate significant visual and stylistic similarities between AI-generations and unauthorized datasets. For example, an AI-generated image of "a vast grassland in the style of Van Gogh's Starry Night" inherently associates with Van Gogh's artistic domain, even without direct replication of the original artwork. Therefore, a new paradigm is needed to emphasize ownership of styles and content for dataset copyright protection.

Several methods have been proposed for personal dataset copyright protection, including Glaze [12], DIAGNOSIS [13], and Luo et al. [14]. Glaze safeguards personal datasets by introducing calculated perturbations to prevent AI style mimicry during fine-tuning, but Bochuan et al. [15] demonstrate these perturbations are vulnerable to adversarial purification. Furthermore, Glaze's approach inherently restricts legitimate training uses. Besides, DIAGNOSIS, which constructs backdoors based on diffusion model memorization, is an approach to copyright protection. However, integrating backdoors into the datasets may introduce new harmful security risks [16]. Meanwhile, Luo et al. use digital watermarking to detect unauthorized usage, but it lacks robustness, as shown in [17].

To address the above problems, we introduce an implicit Zero-Watermarking scheme that focuses on the distinct style and creative essence ingrained within datasets, rather than merely the digital carriers (e.g., digital images). Inspired by recent studies in disentangled representation learning [18, 19, 20, 21, 22] and IP customization [23, 24, 25], we consider that image generation is conceptualized as a regularized entanglement of styles and contents, within the mutually exclusive contraction domains generalized from the anchor of the original samples. Unlike existing methods of embedding **invisible information** into protected datasets, our approach quantizes the domains representing protected style and content representations into **implicit watermarks** to delineate the copyright boundaries.

In this paper, we aim to generate implicit watermarks from the disentangled style domains of protected units, enabling self-generalization and mutual exclusivity. Specifically, we initially employ the style domain encoder to disentangle each protected unit into its style representation, serving as the center anchor points for the contraction domain. Then, we generalize the contraction domain by the dynamic contrastive learning between central samples and boundary samples of the specific protected unit. Finally, the domain achieves the maximum concealed offset of probability distribution through both the injection of identifier $z$ and dynamic contrastive learning, enabling copyright boundary delineation quantized as implicit watermarks. During the verification phase, to address the complex copyright boundaries in image generation with multiple sources of styles and contents, we propose a verification mechanism utilizing the style domain and watermark distribution to tackle hybrid or partial infringements. We highlight our main contributions as follows:

1. We propose a novel watermarking method for dataset copyright protection against unauthorized AI mimicry. To the best of our knowledge, this work is the first study that facilitates the structured delineation of dataset copyright boundaries in the disentangled style domain. Notably, experiments demonstrate that ours accomplish the one-sample verification challenge for copyright ownership of hybrid or partial infringements.

2. We utilize strategies for the self-generalization and mutual exclusivity of $z$-watermarking, breaking away from the traditional methods of embedding invisible information into datasets.

3. To tackle hybrid or partial infringements in image generation with multiple sources of styles and contents, we introduce the concept of watermark distribution to establish a verification mechanism for dataset copyright ownership by the disentangled style domain.

4. Extensive experiments on benchmark datasets demonstrate the effectiveness, robustness, and versatility of our method against various challenges, including adversarial fine-tuning methods (e.g., Dreambooth), watermark removal (e.g., Latent attack) and the usage detection of unauthorized data in black-box cross-APIs and models (e.g., DALL·E·3).

## 2 Related Works

### 2.1 Text-to-Image Generation and Diffusion Models

Recently, the field of visual synthesis has experienced significant advancements, with various research [1, 26, 27, 28, 29, 30, 31, 32] achieving impressive outcomes. Notably, diffusion models [1, 30, 31, 32] have emerged as pioneers in image generation, surpassing earlier models based on adversarial

generative networks [26, 27] and autoregressive methods [28, 29]. Among these, Stable Diffusion [31] stands out for its noteworthy contributions to latent diffusion models. Besides, recent studies, such as Lora [4], Dreambooth [5], Textual inversion [6], and ControlNet [7], have shifted towards personalized fine-tuning of pre-trained diffusion models. These advancements empower individuals to replicate specific styles and contents with just minimal shared unauthorized samples.

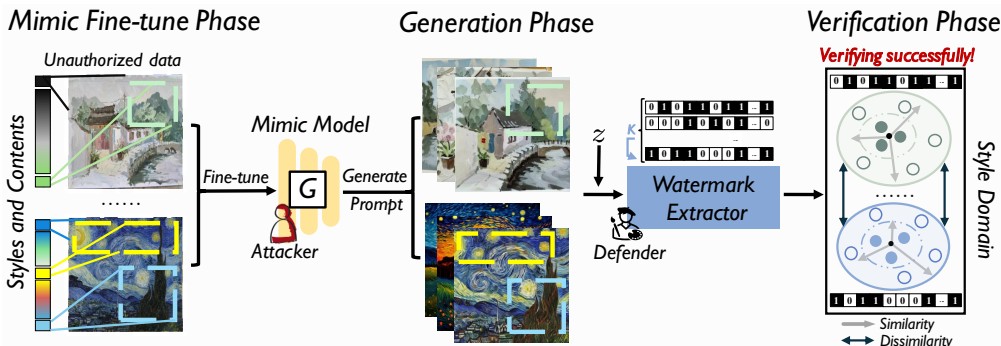

Figure 1: The main pipeline of dataset copyright verification with our method. Notably, we use the watermark extractor (with specific K watermark mapping relationships) and identifier $z$ to detect protected datasets usage, instead of the traditional embedding and extraction pairing process.

## 2.2 Preventing Unauthorized Data Usage

There are several ways [12, 13, 14, 33, 34, 35, 36] to prevent unauthorized data usage. Adversarial example-based methods (i.e., Glaze [12], AdvDM [33], and Anti-Dreambooth [34]) introduce perturbations to induce mimic models to learn different image styles during training and fine-tuning. Nevertheless, the added perturbations are dependent on and constrained by the surrogate model, resulting in weak generalization and transferability. Besides, backdoor-based dataset ownership verification [13, 35, 36] is conducted by defenders triggering whether suspicious models exhibit specific backdoor behaviors. However, the integration of backdoors into datasets could introduce new harmful security risks, as indicated in [16]. At the same time, Luo et al. [14] propose a watermarking framework for detecting art theft mimicry based on digital watermarking techniques [37, 38, 39, 40]. However, the robustness of these is insufficient, as they are easily removable, as indicated in [17].

## 2.3 Disentangled Representation Learning

Disentangled representation learning [41] aims to model the factors driving data variations [42]. Early works [43] used labeled data to factorize representations in a supervised manner. Recently, unsupervised method [44] has been largely explored, especially for disentangling style and content from the image [45, 42, 46, 47, 48]. Inspired by recent studies in disentangled representation learning [18, 19, 20, 21, 22] and IP customization [23, 24, 25], we consider that the act of generating them from scratch requires a deep understanding of the underlying factors and complex generative processes, unlike mere analysis of text or images. In other words, image generation is conceptualized as entangled combinations of styles and contents of the original samples. Taking this viewpoint, we redefine the concept of image beyond digital forms, viewing them as compositions of multiple representations that serve as class-free guidance for diffusion models in self-disentanglement. Additionally, since these disentangled representations are mutually exclusive in high-dimensional space, they naturally demarcate copyright boundaries through mutual exclusivity.

## 3 Method

### 3.1 Threat Model

**Attacker's Goal and Capability.** Attackers could train or fine-tune on protected datasets ($\mathcal{D}$) to replicate the styles and contents of personal portfolios, exploiting them for financial gain or involvement in criminal. The attacker's capabilities are as follows:

- Unauthorized access to proprietary datasets, such as personal portfolios and photo albums.
- Utilize data attacks like second-stage fine-tuning, mixed-clean dilution, purification-latent attack, prompt attack, and data augmentations to remove potentially hidden information.
- Just publicize the APIs and keep the mimicry details hidden, including fine-tuning approaches and training parameters.

**Defender's Goal and Capability.** The defender aims to detect single or minimal instances of mimic images, i.e., publicly available online or offline, to track back copyright ownership. Before sharing data, defenders register the identifier $z$ and implicit watermarks $\mathcal{W}_z$ for protected units with a third party. The defender's capabilities are as follows:

- General Ability: Defenders obtain stylized mimic images from known suspicious models or APIs to verify copyright ownership in black-box setting.
- **Limited Ability:** Defenders occasionally and randomly acquire minimal (even a single) mimic images online or offline without any prior knowledge.

### 3.2 The overview of $z$-Watermarking

The overview of our method is depicted in Figure 2. Our pipeline consists of three phases. Initially, all units within the protected dataset are disentangled, with the contraction domain embedded within the style domain. This is achieved by maximizing the offset via identifier $z$, ensuring non-overlap. Subsequently, self-generalization of each contraction occurs through dynamic contrastive learning between the central and boundary samples of the protected unit. Finally, the watermark is implicitly quantized based on the mutual exclusivity of contraction domains, leveraging their distinct representations of style and content.

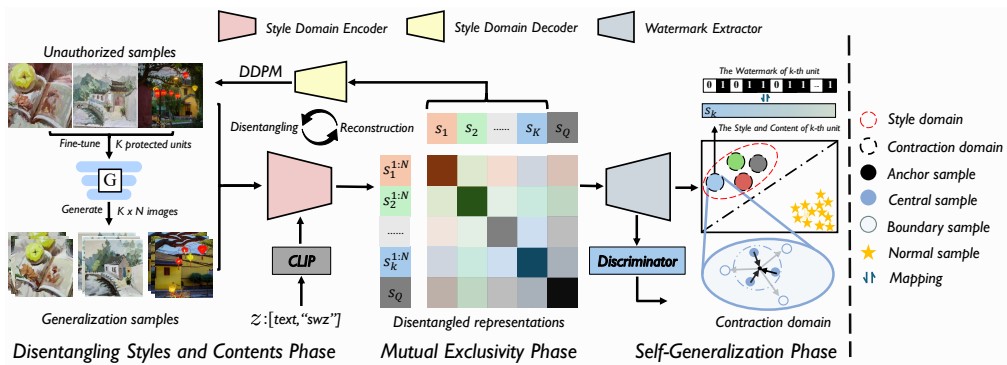

Figure 2: Overview of $z$-watermarking. In this framework, $z$ acts as the key or unique bias of the disentangled style domain $S_z$ of protected units, and $Q$ denotes the dynamic historical negative queue.

### 3.3 Disentangled Style Domain for $z$-Watermarking

The style domain encoder $\mathcal{E}_z$ (ResNet) and denoising decoder $\mathcal{D}_z$ (UNet with $2m+1$ activation layers) are formally defined by a pair of forward and backward Markov chains representing a $T$-steps transformation from a normal distribution $z_T \sim \mathcal{N}(0,1)$ into the learned distribution $z_0 \sim p_\theta(z_x)$. We aim to achieve disentanglement of images at the latent level. To this end, we regularize data $x$ into latent representations $z_x$, which follow a Gaussian distribution $\mathcal{N}$, using a Variational Autoencoder (VAE) as follows.

$$q_\phi(z_x|x) = \mathcal{N}(z_x; \mu(x), \sigma^2(x)I). \tag{1}$$

Eq.1 denotes the probability distribution $q_\phi(z_x|x)$ for $z_x$, which is the mean $\mu(x)$ and variance $\sigma^2(x)$ of $z_x$ (i.e., $\phi$ denotes the parameters of VAE). The style domain encoder is represented as follows: $\mathcal{E}_z(z_x|(x,\phi),z) = s$, where $s = \{v_i\}_{i=1}^m$ (i.e., $s_{1:m}$ is semantically or visually relates to $x$). Identifier $z$ serves as the key or special bias of the style domain $S_z$. Identifier $z$ can be the spatial embedding vector (e.g., image, text, audio, model, etc.). In this paper, we set the text 'swz' to be converted into text feature embeddings by CLIP (i.e., $\phi_z$) as $z$, embedding it into $\mathcal{E}$. Then, we partition the vector $s$

into $m + 1$ sections, which is half the number of layers in the decoder $\mathcal{D}_z$. Each $v_i \in s$ is utilized to modulate the corresponding pair of layers $(h_i, h_{2m-i})$, thus fostering specialization among the latent sub-vectors. Moreover, we implement layer-wise guidance dropout by selectively zeroing out portions of $s_{1:m}$, thereby diminishing the decoder's dependency on sub-vector correlations. The details and tricks are in the supplemental material, and we derive a pre-trained style encoder trained in MS-COCO.

Based on the pre-trained style domain encoder, we design the parameters $\theta_d$ of the discriminator and the $\theta_w$ of watermark extractor. Specifically, let $\mathcal{D} = \{x_i\}_{i=1}^{K}$ that denotes the protected dataset, where $K$ is the number of the protection units (i.e., each sample or class with shared attributes). Let $\mathcal{D}_s = \{x_i^{(n)}\}_{i=1,n=1}^{K,N}$ denotes mimic samples that include $K$ subsets of protected units generated by the surrogate model $\mathcal{M}$ trained on $\mathcal{D}$, where $N$ is the number of mimic samples for the $k$-th protected unit. The optimization objective is as follows:

$$\mathbb{E}_{\mathbb{I}(s_k, s_k^{(n)})} \left[ \mathcal{L}_d((s_k, s_k^{(n)}), z, c_k; \theta_d) + \mathcal{L}_w(s_k, z, w_k; \theta_w) \right]$$

$$s.t. \quad \theta^* = \arg\min_{\theta} \left[ \mathcal{H}_1(\mathcal{C}, \mathcal{D}_s|_{\theta_d}^z) + \mathcal{H}_2(\mathcal{W}, \mathcal{D}_s|_{\theta_w}^z) + \frac{1}{|\mathcal{D}_s|} \sum_{s_k \sim \mathcal{D}}^{K} \sum_{s_k^{(n)} \sim \mathcal{D}_s} (\mathcal{F}_s(s_k, s_k^{(n)}) + \psi) \right],$$

(2)

where $c_k \in \mathcal{C}$ denotes the class of $k$-th protected unit, and $w_k$ denotes the mapping watermark relationship to $k$-th contraction domain. $\mathcal{F}_s$ denotes the cosine similarity function, and $\mathcal{L}(\cdot)$ is the loss function (e.g., $\mathcal{H}_1$ is cross entropy loss, $\mathcal{H}_2$ is mse loss). Specifically, identifier $z$ denotes the representation $s$ is shifted to the marginal distribution. Moreover, $\psi$ in Eq.2 represents the domain regularization term, aimed at achieving dynamic self-generalization and mutual exclusivity of the contraction domain according to the following constraints as Eq.3 and Eq.4.

$$\frac{1}{|\mathcal{D}_s^{k^+}|} \sum_{s_k^+ \sim \mathcal{D}_s^{k^+}} \mathbb{I}(s_k, s_k^+) \le \frac{1}{|\mathcal{D}_s^{k^-}|} \sum_{s_k^- \sim \mathcal{D}_s^{k^-}} \mathbb{I}(s_k, s_k^-) \le c,$$

(3)

Let $x_k \in \mathcal{D}$ and $\{x_k^{(n)}\} = \mathcal{D}_s^k \in \mathcal{D}_s$, where $\mathcal{D}_s^k$ denotes the similar mimic set of the $k$-th protected unit $x_k$. Let $\mathcal{D}_s^k = \mathcal{D}_s^{k^+} + \mathcal{D}_s^{k^-}$, where $x_i^+ \in \mathcal{D}_s^{k^+}$ is the central sample of the contraction domain of $x_k$, and $x_i^- \in \mathcal{D}_s^{k^-}$ is the boundary sample of the contraction domain. $c$ denotes the boundary value of the contraction domain and $\mathbb{I}(\cdot)$ denotes the distance function. We aim for the contraction domain to ensure self-generalization in Eq.3, while evolving mutual exclusivity in Eq.4. Let $x_{\neg k} \in \mathcal{D}_s^{\neg k}$ denote the complement of $\mathcal{D}_s^k$, serving as the negative samples, where $\mathcal{D}_s^{\neg k} = \mathcal{D}_s - \mathcal{D}_s^k$.

$$\prod_{s_k \sim \mathcal{D}_s^k, s_{\neg k} \sim \mathcal{D}_s^{\neg k}} \mathbb{I}(s_k, s_{\neg k}) \gg (c + \beta)^{|\mathcal{D}_s^k| \times |\mathcal{D}_s^{\neg k}|},$$

(4)

where $\beta$ is a positive hyper-parameter. To achieve the above constraint, let $\psi = \lambda_1 \psi_1 + \lambda_2 \psi_2$, where $\lambda_1, \lambda_2$ are two hyper-parameters. $\psi_1$ aim to achieve self-generalization and $\psi_2$ ensures mutual exclusivity and maximum offset described in §3.4 and §3.5.

### 3.4 Self-Generalization Module

As previously mentioned, we aim to explore the style boundaries of the contraction around the anchor sample $x_k$ to establish the range we want to protect. $\psi_1$ aims to achieve self-generalization in Eq.5.

$$\psi_1 = -\log \frac{\exp(s_k \oplus s_i^+/\tau)}{\sum_{i=1}^{N} \exp(s_k \oplus s_i/\tau)},$$

(5)

where $s_k \sim \mathcal{D}$, $s_i \sim \mathcal{D}_s^k$ and $s_i^+ \sim \mathcal{D}_s^{k^+}$. $\tau$ is the temperature parameter. We designate a subset of the $k$-th protected unit as positive samples and the rest as negative (normal). Notably, $s_k$ is disentangled by $\mathcal{E}_z$, while $s_i$ and $s_k^+$ are disentangled by $\mathcal{E}_z'$. Representing the parameters of $\mathcal{E}_z$ as $\theta$ and those of $\mathcal{E}_z'$ as $\theta'$, we update $\theta'$ by momentum update: $\theta' \leftarrow m\theta' + (1 - m)\theta$, where $m$ is a momentum coefficient. Only $\theta$ are updated by back-propagation.

## 3.5 Mutual Exclusivity Module

Let $s_k \sim \mathcal{D}$, $s \sim \mathcal{D}_s$ and $s_i \sim \mathcal{D}_s^k$. The contraction domain of the protected unit is quantized to the predefined implicit watermark via $\mathcal{E}_z$. Formally, the regularization term $\psi_2$ is defined in Eq.6.

$$\psi_2 = -\log \frac{\exp(s_k \oplus s_i / \tau)}{\sum_{s \sim \mathcal{D}_s, q \sim Q} \exp(s_k \oplus (s+q)/\tau)}, \tag{6}$$

where $\mathcal{Q}$ denotes the dynamic historical negative queue. $Q$ is initially populated with anomalous samples and anomaly identifiers, consistently serving as negative instances for contrastive learning. Here, we also employ momentum updates of $\mathcal{E}_z'$ to encode negative examples.

# 4 Dataset Copyright Verification via $z$-Watermarking

**One-Sample Verification.** We aim to verify copyright ownership using single or minimal mimic images. To achieve this, we propose leveraging the disentangled style domain to facilitate the structured delineation of dataset copyright boundaries. Experiments show that our method offers effective copyright verification, with the single-sample success rate far exceeding the baseline, in Table 2. The probability of the watermark guided by $z$ in the contraction domain of the protected unit is denoted as Eq.7.

$$P_z(x|\phi \leftrightharpoons \mathcal{D}) = \frac{q_{\phi_z}(z_{emb}|z)}{2^L \cdot K \cdot (c+\beta)^{K \times N^2 \times (K-1)}}, \tag{7}$$

where $\lim_{x \to \mathcal{D}} P_z(x|\phi) = \eta$ indicates $x$ (mimic sample) originating from protected $\mathcal{D}$ is an extremely unlikely probability event (i.e., $\eta$ denotes infinitesimal). The occurrence of extremely low-probability events ensures the credible mathematical basis for the ownership of sample copyrights. Meanwhile, in generative scenarios, the challenge of identifying the target carrier ( relying solely on manual similarity judgments) renders traditional one-to-one watermark verification mechanisms limited.

**Extensive Statistical Verification.** To further validate our method's effectiveness from multiple perspectives, we propose the concept of watermark distribution. Assuming the multi-styles and multi-contents of $T$ datasets are present in $\mathcal{D}_m$, $T$ defenders utilize $\{\mathcal{E}_{z_t}, z_t\}_{t=1}^T$ to disentangle the image generations. Let $\mathcal{E}_{z_t}(x) = (c_t, w_t)$, where $c_t \in \{0, k_t\}$ and $k_t \in K_t$ (i.e., $K_t$ is the number of protected units of $t$-$th$ dataset, and the corresponding watermark is $w_{k_t}$). Watermark distribution is defined as Eq.8,

$$t@wd = \frac{\sum_{(c_t, w_t) \overset{\varepsilon_{z_t}}{\sim} \mathcal{D}_m} \{\mathcal{F}_b(w_t, w_{k_t})\}_{c_t = k_t}}{|\mathcal{D}_m|}. \tag{8}$$

Let $t@k@100\%wd = t@wd[\arg max_{k_t} \{\mathcal{F}_b(\cdot) = 100\%, c_t = k_t\}]$ that denotes the best distribution (i.e., within $\mathcal{D}_m$, the number of samples of the $k_t$ type of $t$-$th$ datasets is the largest) in the most accurate distribution (i.e., with bits accuracy reaching 100%). When assessing data copyright for single-party verification, two criteria must be fulfilled: $Avg\ acc > \alpha$ and $t@k@100\%wd > \gamma$, where $Avg\ acc$ represents Average Watermark Accuracy. In this paper, the threshold for $\alpha$ is set to 0.99, and the threshold for $\gamma$ is set to 0.80. For multi-party hybrid or partial infringements, copyright ownership is determined by comparing the maximum value of $t@k@100\%wd$ tested across $T$ different style domain encoders in $\mathcal{D}_m$, attributing it to the $k$-$th$ unit of the $t$-$th$ protected dataset.

# 5 Experiments

## 5.1 Experimental Setting

**Datasets and Models.** In this paper, we pre-train the style domain encoder [49] and decoder [50] on MS COCO [51]. We conduct experiments on three open-source benchmark datasets (i.e., CelebA [52], Pokenmon [53], Dreambooth dataset [5]), 17 Artists (e.g, Van Gogh and Monet) and 10 AI' artworks (e.g., GhostMix and CatLora). The surrogate model is Stable diffusion v1.5 [32] fine-tuned (i.e., Lora [4] and Dreambooth [5]) on the benchmark datasets. Moreover, the attackers include Stable Diffusion v1.5&v2.0, and the APIs of DALL·E-3 [3], Imagen2 [1], PG-$v$2.5 [54], PixArt-$\alpha$ [55]. We use CLIP to extract the $z_{emb}$ of text-$z$ in our setting. Specifically, we randomly select a subset containing $K$ protected units from each dataset for training (i.e., $N$ mimic images per unit generated).

**Baseline and Metrics.** We compare our pipeline against existing digital watermarking methods (i.e., DCT-DWT-SVD [39], RivaGan [37], SSL [40], Trustmark [56]), and RoSteALS [57]). We evaluate each method's performance using average watermark accuracy ($Avg\ acc$) and watermark distribution metrics ($t@wd$ and $t@k@100\%wd$, as defined in §4). Additionally, we employ FID and CLIP Score to evaluate the quality of AI-generation, and True Positive and True Negative to measure Discriminator performance.

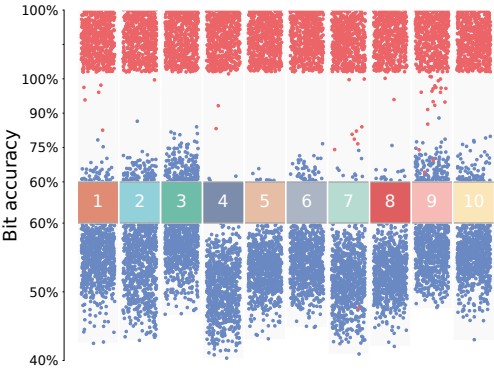

(a) The watermark distribution across generated samples from 10 random protected units.

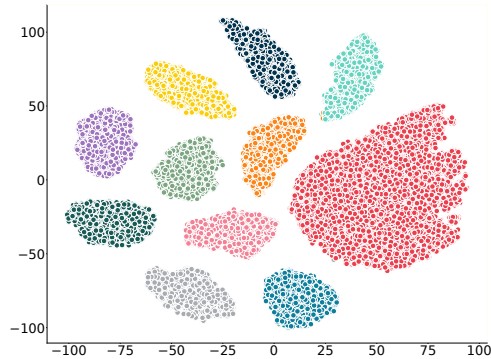

(b) The t-SNE of feature representations for mimic samples from random protected units in Artists.

Figure 3: Main results for our method. The left subfigure compares watermark distribution between mimic (Red) and no-mimic (Blue) models from 10 random protected datasets. The right subfigure illustrates the structured delineation of style domains' boundaries.

## 5.2 Main Result

To benchmark the effectiveness of the watermark, we primarily report the watermark distribution across 1000 image generations from all units of each protected dataset in the black-box validation scenario of AI mimicry, utilizing $Avg\ acc$ and $t@k@100\%wd$ (i.e., the proportion of samples where watermark bit accuracy hits 100%) for evaluation.

Table 1: Main results. We enumerate the sample count within each range of watermark distribution (128 bits) from 1000 mimic images of all protected units for both mimic and non-mimic models.

| 128-bit | w\mimicry | The sample counts within each range of watermark distribution | | | | | | $Avg\ acc$(%) | $t@k@100\%wd$ (%) |
| --- | --- | --- | --- | --- | --- | --- | --- | --- | --- |
| | | 0-20% | 20-40% | 40-60% | 60-80% | 80-90% | 90-100% | | |
| CelebA | ✗ | 0 | 272 | 495 | 231 | 2 | 0 | 51.46 | 0 |
| | ✓ | 0 | 0 | 1 | 1 | 3 | 995 | **99.81** | **98.1** |
| CUB | ✗ | 0 | 249 | 513 | 227 | 11 | 0 | 49.26 | 0 |
| | ✓ | 0 | 0 | 2 | 5 | 5 | 988 | **99.56** | **96.4** |
| Dreambooth | ✗ | 0 | 127 | 524 | 341 | 8 | 0 | 55.71 | 0 |
| | ✓ | 0 | 0 | 0 | 1 | 3 | 996 | **99.97** | **98.1** |
| Artists | ✗ | 0 | 124 | 455 | 419 | 2 | 0 | 55.83 | 0 |
| | ✓ | 0 | 0 | 0 | 1 | 6 | 993 | **99.87** | **97.9** |
| AIs | ✗ | 0 | 144 | 561 | 292 | 3 | 0 | 52.20 | 0 |
| | ✓ | 0 | 0 | 0 | 1 | 1 | 998 | **99.95** | **98.0** |

Main experimental findings regarding watermark distribution validation on both mimic and no-mimic models across five datasets are detailed in Figure 3a and Table 1. Our method outperforms the watermark distribution under random states, which are not exposed on the protected dataset. Specifically, our average accuracy exceeds 99%, significantly higher than the approximately 50% of the non-infringement state model. Such a significant difference in watermark distribution is one of the key pieces of evidence for copyright authentication. Additionally, we compare our method with digital watermarking approaches in Table 2, where our method achieves an $Avg\ acc$ of 99.83%, surpassing others that reach only around 60%. Notably, in cases of 100% watermark accuracy ($t@k@100\%wd$), the proportion of samples using digital watermarking methods is deficient, whereas

our method reaches 97.7%. This indicates the failure of digital watermarking in attributing ownership in AI mimicry scenarios, contrasting with the effectiveness of our proposed method.

Table 2: Main results. Comparison of results across different watermarking methods. The stark disparity in $t@k@100\%wd$ between our results and the baseline reveals that traditional invisible watermarks are prone to be removed or diluted during diffusion training.

| Method | $Avg\ acc$ (%) ↑ | $t@k@100\%wd$ (%) ↑ |
|---|---|---|
| DCT-DWT-SVD | 57.76 | $\leq 0.1$ |
| RivaGan | 61.34 | $\leq 0.1$ |
| SSL | 64.39 | $\leq 0.1$ |
| Trustmark | 55.37 | 6.6 |
| RoSteALS | 66.50 | 7.9 |
| Ours | **99.83** | **97.7** |

## 5.3 Robustness Study

To benchmark the robustness of our watermark, we document its performance against various attack methods. These include identifier $z$ error, second-stage fine-tuning with a 1:10 ratio between original and generated images, mixed clean fine-tuning with a blending rate of 0.1, watermark removal in latent attack [17], and prompt attack with different description in black-box scenarios. Additionally, we utilize 7 data augmentations as attacks, consisting of 90° rotation, 50% JPEG compression, 60% center cropping and scaling, Gaussian blur with a 3×3 filter size, color jitter with a hue factor of 100, along with adjustments to brightness by a factor of 1.5 and contrast by a factor of 2.0.

Table 3: The results of the robustness study. We conduct robustness experiments from various attack perspectives, including identifier $z$ error, second-stage fine-tuning, mixed clean fine-tuning, watermark removal in latent attack, prompt attack, and image augmentations.

| 128-bit | The sample counts within each range of watermark distribution | | | | | | FID | CLIP | $Avg\ acc$ (%) ↓ | $t@k@100\%wd$ (%) ↓ |
|---|---|---|---|---|---|---|---|---|---|---|
| | 0-20% | 20-40% | 40-60% | 60-80% | 80-90% | 90-100% | | | | |
| w\o mimicry | 0 | 124 | 455 | 419 | 2 | 0 | - | - | 55.71 | 0 |
| w\o correct $z$ | 0 | 151 | 555 | 291 | 3 | 0 | - | - | 52.15 | 0 |
| w\o Attack | 0 | 0 | 0 | 1 | 6 | 993 | 266.48 | **0.9491** | 99.87 | **97.9** |
| Second-stage Fine-tune | 0 | 0 | 6 | 9 | 7 | 975 | 271.54 | 0.9358 | 99.13 | 93.3 |
| Mixed Clean Fine-tune | 0 | 1 | 11 | 29 | 35 | 944 | **259.89** | 0.9337 | 99.04 | 92.2 |
| Latent Attack | 0 | 0 | 13 | 19 | 24 | 925 | 289.75 | 0.9094 | 95.81 | 87.2 |
| Prompt Attack | 0 | 0 | 95 | 9 | 36 | 860 | 310.68 | 0.9094 | 95.81 | 76.7 |
| Contrast | 0 | 0 | 8 | 9 | 11 | 972 | 318.39 | 0.8951 | 99.01 | 92.2 |
| JPEG | 0 | 0 | 8 | 10 | 14 | 968 | 307.41 | 0.8399 | 98.97 | 91.6 |
| GaussianBlur | 0 | 0 | 11 | 17 | 15 | 957 | 341.04 | 0.9017 | 98.50 | 89.8 |
| Brightness | 0 | 0 | 24 | 22 | 19 | 935 | 318.41 | 0.8839 | 97.63 | 88.1 |
| CenterCrop | 0 | 0 | 43 | 82 | 68 | 805 | 379.10 | 0.8216 | 94.82 | 69.9 |
| Hue | 0 | 0 | 37 | 80 | 50 | 833 | 339.76 | 0.8362 | 94.44 | 68.6 |
| Rotation | 0 | 17 | 294 | 415 | 105 | 169 | 394.54 | 0.8124 | 83.66 | 14.8 |

In Table 2, baseline methods fall short in attributing copyright ownership due to their inability to extract the complete watermark, achieving a likelihood of less than 0.1. Conversely, our methods demonstrate heightened reliability. Table 3 reveals that even under adversarial conditions, we can extract numerous samples with 100% bit accuracy, surpassing the performance of baseline models (i.e., $t@k@100\%wd$ less than 0.1%). Notably, while attacks decrease $t@k@100\%wd$, only a certain proportion of $t@k@100\%wd$ samples is required for verification in AI mimicry copyright attribution. This is attributed to the improbable occurrence of such events in a natural state, as outlined in Eq.7.

## 5.4 Generalization Study

To benchmark the generalization capability of our watermark, we document its performance in copyright verification within the landscape of AI mimicry, considering an array of fine-tuning models and black-box APIs in Table 4. We set the surrogate model to Stable Diffusion v1.5. Our primary focus lies on examining its generalization across Stable Diffusion v1.5 (i.e., Lora and Dreambooth) & v2.0, as well as the APIs of PixArt-$\alpha$, PG-$v2.5$, DALL·E·3, and Imagen2. Each model and API is instructed to generate 20 images for inspection using the prompt "An art piece resembling the style of 'Starry Night'", aiming to discern whether the attack model has been exposed to Van Gogh's

portfolios. In Table 4, our method achieves 100% *Avg acc* on most suspicious models, with an overall average of 98.60%. The $t@k@100\%wd$ also reaches nearly 100% on most models, with an average level of 94.29%. Additionally, it achieves a 100% accuracy in True Positive (TP) detection, as well as 100% *Avg acc* and $t@k@100\%wd$. As depicted in Figure 4, our proposed method provides strong evidence of its ability to detect imitation behavior of commercial APIs like PixArt-$\alpha$ using a few suspicious samples.

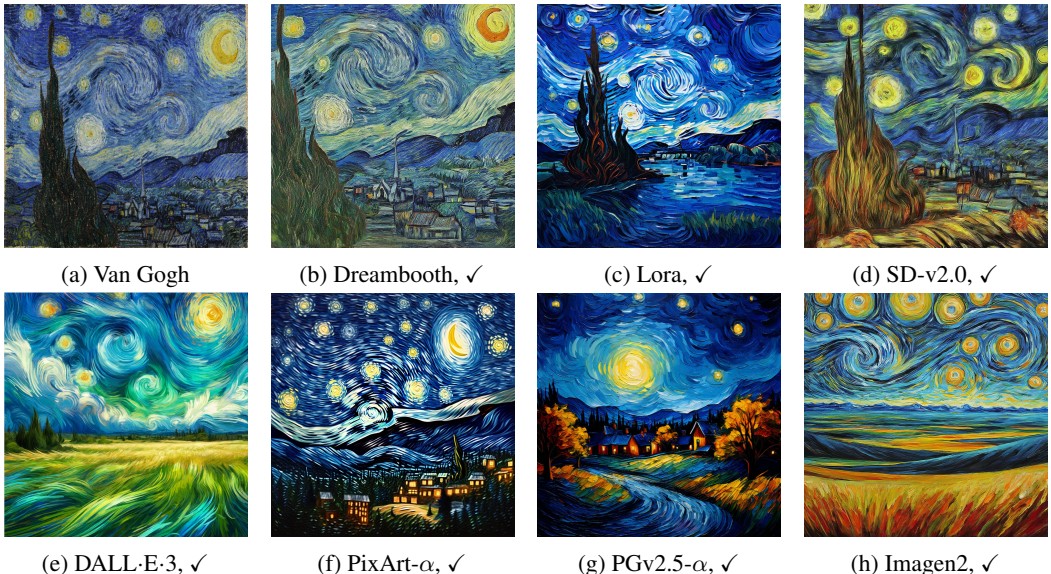

| (a) Van Gogh | (b) Dreambooth, ✓ | (c) Lora, ✓ | (d) SD-v2.0, ✓ |

| (e) DALL·E·3, ✓ | (f) PixArt-$\alpha$, ✓ | (g) PGv2.5-$\alpha$, ✓ | (h) Imagen2, ✓ |

Figure 4: The results of generalization study. Each API or model is instructed to generate the image for data copyright ownership using the prompt "An art piece resembling the style of 'Starry Night'". Among them, subfigure 4a represents the protected sample units, while subfigures 4b-4h, represent the suspicious mimic samples generated by various suspicious models.

Table 4: The results of generalization study. Utilizing the prompt "An art piece resembling the style of 'Starry Night'", we generate 20 images by suspicious models and APIs of black-box.

| Attacker models/APIs | FID | CLIP | TP | TN | *Avg acc* (%) | $t@k@100\%wd$ (%) |
|---|---|---|---|---|---|---|
| SD-v1.5 + Dreambooth | 259.76 | 0.9484 | 20 | 0 | 100 | 100 |
| SD-v1.5 + Lora | 265.21 | 0.9396 | 20 | 0 | 100 | 100 |
| SD-v2.0 | 267.38 | 0.9163 | 20 | 0 | 100 | 100 |
| PixArt-$\alpha$ | 285.09 | 0.9011 | 20 | 0 | 100 | 100 |
| PGv2.5 | 301.94 | 0.8836 | 19 | 1 | 98.98 | 95.0 |
| DALL·E·3 | 318.13 | 0.8966 | 18 | 2 | 97.02 | 85.0 |
| Imagen2 | 326.09 | 0.9368 | 17 | 3 | 94.35 | 80.0 |
| Average | 289.09 | 0.9175 | - | - | 98.60 | 94.29 |

## 5.5 Ablation Study

We hereby discuss the effects of several key hyperparameters involved in $z$-watermarking. Please find more experiments regarding other parameters and detailed settings in the Appendix.

**Model Component Study.** We evaluate the effectiveness of our components: Disentangled Style Domain D, self-generalization module $\psi_1$, and mutual exclusivity module $\psi_2$. As shown in Figure 5a and 5d, we compare *Avg acc* and $t@k@100\%wd$ across five datasets. It demonstrates that omitting any proposed components leads to a decline in the model's performance. Notably, the Disentangled Style Domain enables the $t@k@100\%wd$ to rise from around 60% to over 90%, while the addition of $\psi_1$ and $\psi_2$ further optimizes the model to achieve performance exceeding 99%.

**Data Scale Study.** We next explore the relationship between the training data scale and validation data scale in Figure 5b and 5e, which is crucial for real-world copyright protection scenarios. Experimental

evidence suggests that a minimal set can propel our model to an above 99% $Avg\ acc$. Besides, high $t@k@100\%wd$ is observed on a limited scale of validation dataset, reflecting the effectiveness of practical validation settings, where access to the APIs of suspicious black-box mimic models.

**Bit Length Study.** Watermark bit length serves as the bedrock of copyright verification reliability theory in AI mimicry scenarios. Figure 5c and 5f present experiments validating lengths from 32 to 512 bits across five datasets. Notably, experiments with a bit length of 512 demonstrated higher average accuracy and $t@k@100\%wd$, highlighting the scalability and reliability of $z$-watermarking.

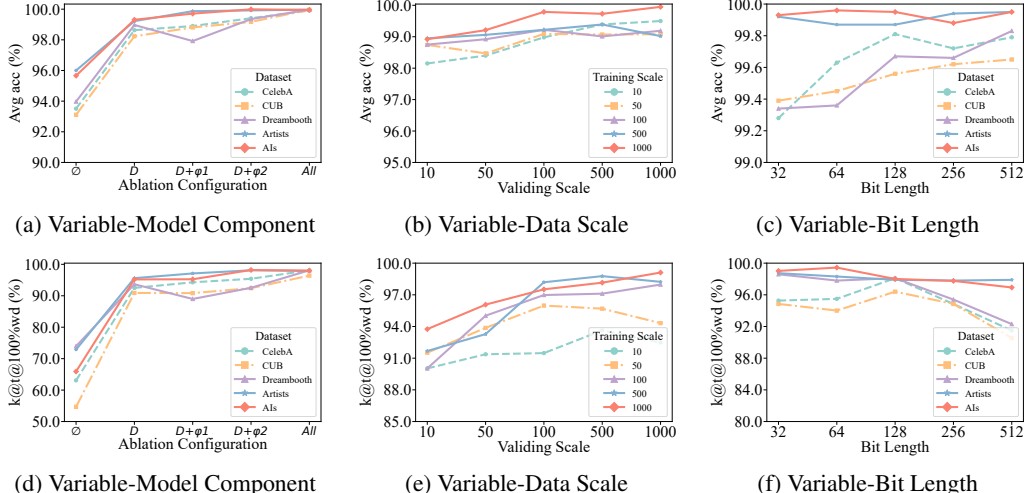

Figure 5: The results of Ablation study. We performed ablation experiments mainly on model components (Distangled style domain D, self-generalization $\psi_1$, and mutual exclusivity $\psi_2$), data scale (ranging from 10 to 1000), and watermark bit length (ranging from 32 to 512).

# 6 Discussion and Limitation

**Ethical Statement:** To enrich our experimental dataset, we gathered a lot of contemporary authorized artworks, historical artworks, and AI-generated pieces. We at this moment affirm that this collected data is specifically for the experiments related to our method and is not for any other use.

**Limitation:** Our proposed method focuses on the disentangled style domains of protected units. Consequently, modifications to the deep features of style domains using existing techniques (e.g., style transfer or bias injection), are expected to influence our performance. Although our robustness study confirms our reliability against various attacks, the potential impact of specific targeted attacks (e.g., Glaze) on performance degradation could not be overlooked. To address this challenge, adding specific adversarial samples for adversarial optimization could guide our future research. Additionally, the fine-tuning performance of the surrogate model may slightly affect the experimental results, so it is necessary to set the optimization parameters appropriately.

# 7 Conclusion

This paper presents the first study on the disentangled style domain for implicit watermarking to detect unauthorized data usage of AI mimicry, from the perspective of entity protection in styles and contents. Extensive experiments demonstrate the superiority of $z$-watermarking compared to the baseline. Notably, our method achieves one-sample verification for copyright ownership of hybrid or partial AI infringements. We aspire for our work to advance the ethical evolution of future artificial intelligence, ensuring due respect for creators' copyrights.

# Acknowledgements

This work was supported by the National Key R&D Program of China under Grant 2023YFF0905000.

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
