# OpenReview forum: "Disentangled Style Domain for Implicit $z$-Watermark Towards Copyright Protection"
_NeurIPS.cc/2024/Conference — NeurIPS 2024 poster_

### Official Review · Reviewer_NU2V · 2024-07-11

**Soundness:** 2
**Presentation:** 2
**Contribution:** 2
**Rating:** 4
**Confidence:** 4

**Summary:**

The current watermarks applied to AI-generated images rely on adding additional information, limiting their ability to detect unauthorized use of data. This paper introduces a new implicit watermarking scheme, which first utilizes the disentangled style domain to detect unauthorized dataset usage in text-to-image models., so as to achieve self-generalization and mutual exclusivity within the style domain anchored by protected units. In addition, this paper introduces the concept of watermarking distribution and establishes a verification mechanism for copyright ownership of hybrid or partial infringements. It is worth noting that this paper implements One-Sample-Verification for dataset copyright verification in AI mimic generation. This paper also designed experiments to verify the robustness, generalization and ablation of multiple data sets.

**Strengths:**

- The problem exploited in this paper is important and needs to be addressed.
- The idea of disentangling the image into content and style seems effective.

**Weaknesses:**

- This paper is really hard to follow. The method's modules are not explained very well, and the input, output and training details of each module are not clear.
- Too much mathematical description makes it difficult for readers to quickly understand the process and principle of the method, and the meaning of each mathematical symbol is not clearly described.
- The model structure diagram is less relevant to subsequent descriptions. The paper also lacks simple examples of the data used in the experiment.
-Only the watermark distribution concept is proposed, and the copyright ownership verification experiments of hybrid or partial infringements are lacking.

**Questions:**

- The significant problem is the writing, too obscure.
- There is a problem with the icon marking in Figure 2, the marked color of central sample is inconsistent with that in the figure.
- There is an unclear sign in line 181. Should $s_k^+$ be $s_i^+$?
- Can you explain the meaning of each symbol of formula 7 in detail, and supplement the rationality and significance of the indicators proposed?
- In 5.2 Main result, can you give the value of the number of protected units (i.e., K) and how the 1000 images used were selected?
- It is observed that the avg acc is higher with the longer the watermark length in the ablation experiment. Can you give the details of the mapping from the contraction domain to the watermark in the extractor?
- The experiment is only compared with the digital watermarking method, can you add a comparison with other methods (such as backdoor-based)?

**Limitations:**

The authors addressed the limitations of this paper, which focuses only on the disentangled style domains of protected units, potentially making it difficult to resist attacks that modify deep features of style domains. Future research could incorporate specific adversarial samples.

---

> ### Author Rebuttal · Authors · 2024-08-06
>
> Dear Reviewer NU2V, thank you very much for your careful review of our paper and thoughtful comments. We hope the following responses can alleviate your concerns.
>
> ---
>
> **Q1: The significant problem is the writing, this paper is tough to follow and not examples.**
>
> **R1:** Thank you for your constructive suggestions! We will reorganize and improve it to make the expression clearer and more understandable. **Regarding training details**, we have provided **model details and experiment details** in the supplementary materials. Meanwhile, **in Section 3.2 of the supplementary materials**, we show simple examples of the suspicious data generated from suspicious models and APIs, such as DALL·E·3.
>
> ---
>
> **Q2: Can you explain the meaning of each symbol of Formula 7 in detail, and supplement the rationality and significance of the indicators proposed?**
>
> **R2:** Formula 7 is as follows:
> $P_{z}(x|\phi \backsimeq \mathcal{D}) = \frac{q_{\phi_z}(z_{emb}|z)}{2^L \cdot K \cdot (c+\beta)^{K \times N^2 \times (K-1)}}$
> - **On the left side of the equation:** $x$ denotes the suspicious sample, $\phi$ represents the parameters of the VAE , $\mathcal{D}$ denotes the protected dataset, $z$ signifies the identifier. And $p(\cdot)$ represents the probability distribution of the copyright of $x$ belonging to $\mathcal{D}$.
> - **On the right side of the equation:** $z_{emb}$ denotes the embedding representation, and $q_{\phi_z}(z_{emb}|z)$ denotes the prior probability distribution. $L$ denotes the length of the watermark, $K$ denotes the number of protected units, $N$ denotes the number of generation of protected unit, $c$ denotes the marginal distance, and $\beta$ denotes a positive hyper-parameter.
>
> Here, $/2^L$ denotes the probability of the watermark conforming to $L_{bit}$, $1/K$ denotes the probability that the sample to be detected belongs to $K$ datasets' class, and $1/(c+\beta)^{K\times N^2\times(K-1)}$ denotes the reciprocal of the distance between samples with different styles and contents. Their product represents the probability that the sample to be detected originates from the protected dataset. In hypothesis testing, a low-probability event is almost unlikely to occur in a single random trial, and the probability of such an event is used as the significance level $\alpha$ (i.e., $\alpha$ $ \leq P(\cdot))$. Therefore, in the process of copyright ownership detection, the event that the sample is detected as belonging to the protected dataset can be expressed as $H_0: D \leftarrow x$, with a confidence interval of $1 - \alpha\$, and can be expressed as $P(|X - \mathcal{D}| \leq c) = 1 - \alpha$. Thus, we have a very high confidence in ensuring the ownership.
>
> ---
>
> **Q3: Can you give the value of the number of protected units (i.e., K) and how the 1000 images used were selected?**
> **R3:** Thank you for your comments and we do understand your concerns.
> 1. In 5.2 Main result, we set the value of $K$ to 50.
> 2. In the process of selecting 1000 images of the protected unit, we first obtained the representation $z$ of each image through the style domain encoder. We randomly selected one as the $z_o$ anchor sample, and the others as $z_{go}$. Then, we ranked them based on their similarity and Euclidean distance and finally selected the images according to the ranking results.
>
> **Q4: Can you give the details of the mapping from the contraction domain to the watermark in the extractor?**
>
> **R4:** Thank you for your comments. The PyTorch code of the watermark extractor is as follows:
>
> ```
> class w_decoder(nn.Module):
>     def __init__(self, inc, outc):
>         super(w_decoder, self).__init__()
>         self.Conv_ = nn.Sequential(
>             nn.Conv2d(4*inc, 4*inc, 3, 2, 1),
>             nn.BatchNorm2d(4*inc),
>             nn.GELU(),
>             nn.Conv2d(4*inc, 4*inc, 3, 2, 1),
>             nn.BatchNorm2d(4*inc),
>             nn.GELU(),)
>         self.F_fusion = nn.Sequential(
>             nn.BatchNorm1d(6*inc),
>             nn.GELU(),
>             nn.Linear(6*inc, inc),
>             nn.BatchNorm1d(inc),
>             nn.GELU())
>         self.F_reduce = nn.Sequential(
>             nn.Linear(4*inc, 2*inc),
>             nn.BatchNorm1d(2*inc))
>         self.out = nn.Sequential(
>             nn.BatchNorm1d(inc),
>             nn.GELU(),
>             nn.Linear(inc, outc))
>         self.fc_d = nn.Linear(inc, inc)
>         self.fc_f = nn.Linear(inc, inc)
>         self.Adapt = nn.AdaptiveAvgPool2d(1)
>     def forward(self, data, domain, z=None):
>         f = self.Conv_(domain)
>         f = self.Adapt(f).view(f.shape[0], f.shape[1])
>         f_reduce = torch.cat((self.F_reduce(f), f), dim=-1)
>         f_fusion = self.F_fusion(torch.add(f_reduce, z))
>         out = self.fc_d(data) + self.fc_f(f_fusion) + data + f_fusion
>         out = self.out(out)
>         return out
> wm_logits = Z_Model.w_decoder(data, domain, z)
> ```
> ---
> **Q5: Can you add a comparison with other methods (such as backdoor-based)?**
>
> **R5:** To further alleviate your concerns, we compare ours and methods based on backdoor attacks.
> - We employ the current SOTA (DIAGNOSIS[1]) for dataset protection through backdoor. The evaluation metrics utilized are True Positive (TP), True Negative (TN), and Attack Success Rate (ASR), as implemented by DIAGNOSIS.
> - **Main result:** The experimental results are shown in the table below.
>
> |Method|TP|TN|ASR(%)|Avg acc(%)|k@t@100%wd(%)|
> |-----------|-----|-----|---------|-------------|-------------------|
> |DIAGNOSIS|993|7|99.3|-|-|
> |Ours|1000|0|100|99.72|98|
>
> - **Post-tracking ownership:** It refers to the process of claiming copyright ownership before litigation when owners discover suspicious models or images without having embedded backdoors immediately.
>
> |Method|TP|TN|ASR(%)|Avg acc(%)|k@t@100\%wd(%)|
> |-----------|-----|-----|---------|-------------|-------------------|
> |DIAGNOSIS|2|998|0.2|-|-|
> |Ours|1000|0|100|99.69|94.7|
>
> [1] DIAGNOSIS: Detecting Unauthorized Data Usages in Text-to-image Diffusion Models. ICLR 2024.

---

> > ### Comment · Reviewer_NU2V · 2024-08-11
> > **Comment**
> >
> > Thank you for your response. These details are important so the reader can understand your method clearly, I will raise my score. However, after reading other reviewer's comments, I also have the same concern: the efficiency of this proposed framework should also be analyzed.

---

> ### Author Response · Authors · 2024-08-12
>
> Dear Reviewer **NU2V**, thank you once again for your valuable feedback on our paper and for helping us improve our work. Your decision to raise our score is a recognition of our efforts. Regarding your concerns about the efficiency of the proposed framework, we have addressed this in our response to Reviewer d18z by analyzing the pipeline of our method across _the registration, computation, and inference stages_, supported by detailed experimental data. We will further clarify the efficiency of this framework in the revised paper. Thank you again for your thorough review and for helping us improve our work!

---

### Official Review · Reviewer_d18z · 2024-07-12

**Soundness:** 2
**Presentation:** 2
**Contribution:** 2
**Rating:** 4
**Confidence:** 3

**Summary:**

The paper introduces an implicit watermarking scheme that leverages disentangled style domains to detect unauthorized dataset usage in text-to-image models. The proposed method aims to address limitations in traditional watermarking techniques by using implicit z-watermarks for dataset copyright verification, achieving better robustness against various attacks.

**Strengths:**

The proposed framework is able to protect the dataset copyright. This is extremely important in an era of generative AI. Besides, rather than directly protecting the image, this work proposes to protect the styles. Such a thinking may bring some new insights into this area.

**Weaknesses:**

1. The writing of this paper has some problems. The description of its first section is totally in a chaos status.
2. In some specific situations, I agree that the styles are necessary to be protected. However, in most situations, copyright law typically protects original works of authorship. An individual style, which may consist of a particular technique or aesthetic, is not considered a tangible, original creation. Styles are more akin to ideas or methods, which are generally not protected under copyright law. If the styles are not protected by the laws, why do we need such a method mentioned in this submission?
3. Defining and enforcing copyrights for individual styles would be highly subjective and impractical. Styles evolve and are influenced by many sources, making it difficult to establish clear boundaries for what constitutes a protected style.
4. Granting copyright protection to individual styles could stifle creativity and innovation. Artists and creators often build upon existing styles and techniques to develop new works. Restricting the use of styles could hinder the creative process and limit artistic freedom. For example, if the Cubist style were copyrighted, it could prevent new artists from experimenting with and developing this style further, thereby limiting artistic progress.

**Questions:**

I have shown my concerns in the weakness part, Please address my concerns there.

**Limitations:**

I am uncertain about how practical it is to implement this method on a large scale. The experiments provided in this paper only cover limited data.

---

> ### Author Rebuttal · Authors · 2024-08-06
>
> Dear Reviewer **d18z**, thank you very much for your careful review of our paper and thoughtful comments.
>
> **Q1: Response on "Why We Need to Protect Individual Creators' Styles and Content Copyrights in the AI Era".**
>
> **R1:** To further alleviate your concerns, we provide more explanations.
> - First, **considering both style and content align better with judicial standards**. In copyright cases, judges assess ownership by comparing the style, brushstrokes, and content of the original and imitation works.
>
> - Second, **there have been numerous high-profile legal cases involving unauthorized imitation of artists' styles by AGI.** Notable Sarah Andersen [1] and Getty Images [2] have filed lawsuits against Stability AI, DeviantArt, Midjourney, and OpenAI over copyright and trademark infringement. The AI-generation 'Rock, Paper, Scissors'[3] has violated multiple laws, including the Digital Millennium Copyright Act. Greg Rutkowski’s art style has been used by AI without authorization for profit over 3 million times [4]. _In the AIGC era, safeguarding and tracing the styles and content of personal works is crucial_.
> - Third, **there are already some efforts dedicated to protecting the styles of artistic works,** such as **Glaze: _Protecting Artists from Style Mimicry by Text-to-Image Models_[5] (_Best Paper at USENIX Security 2023_).** Glaze's survey shows that 91% of 1,207 artists are concerned about AI using their works for training. **Artists expect AI mimicry to have a significant impact on the art community:** **97% of artists believe it will reduce job security for some artists; 88% think it will discourage new students from studying art; and 70% feel it will diminish creativity.** Many artists (**over 89%**) have already taken or plan to take action in response to AI mimicry. Additionally, **55%** think reducing their online presence will impact their careers, while **78%** of artists expect AI mimicry to affect job security, rising to **94%** for newer artists. The survey report indicates that without appropriate regulations, AI-style mimicry could undermine people’s motivation for creative freedom.
> - Fourth, **we aim to establish a positive and healthy cycle for the development of art between generative AI and human creation,** where human creators should retain the rights to authorize and trace their creative styles and content. We hope our work will guide the regulated development of generative AI. _In the AI era, we believe this step is urgent for ensuring intellectual property rights for human creativity._
>
> **Q2: Defining and enforcing copyrights for individual styles would be highly subjective and impractical. Styles evolve and are influenced by many sources, making it difficult to establish clear boundaries for what constitutes a protected style.**
>
> **R2: From a computational perspective, the abstract high-dimensional features of the style domain have distinctiveness.** In this paper,**to ensure the exclusivity of the style domain**, we use identifier $z$ to maximally shift the contraction domain to the edge distribution of the style representation space. _Specifically, We decouple the style domain and perform dynamic contrastive learning to increase the similarity distance._ **The Style Domain is shifted into the contraction domain of the edge distribution by $z$; $z$ and the watermark to be verified are held by the defender.** The copyright boundary of the protected unit, which is the edge space distribution, can only be correctly tracked and yield the correct watermark when using $z$.
>
> ||TP|TN|Avg acc(%)|k@t@100%wd(%)|
> |-------------------|-----|-----|-----------------|--------------------|
> |$z$ Error|0|1000|52.15|0|
> |$z$-watermarking|1000|0|99.87|97.9|
>
> **Q3: Regarding the statement ‘Granting copyright protection to individual styles could hinder creativity and artistic progress by restricting the use and evolution of existing styles.’**
>
> **R3:** We do understand your concerns. Next, we will provide a more detailed explanation below.
>
> - **First,** **we should clarify our goal again: creators should have the right to authorize and trace their creative styles and content,** particularly in the context of unauthorized mimicry by generative AI. In the AI era, we believe this step may be urgent: ensuring intellectual property protection for human creativity.
> - **Second, current instances of AI mimicry may severely hinder their motivation and damage their enthusiasm, turning high-quality works into someone else’s benefit according to Glaze's survey [5].** The survey of **1,207 artists** shows **91%** are worried about AI training on their works. Concerns include job security (**97%**), deterring new students (**88%**), and reduced creativity (**70%**). **89%** are taking action, with **53%** considering reducing their online presence. **77%** believe AI mimics their styles well, but unauthorized use remains a major concern. _Glaze collaborates with **art-centric social networks**, advocacy groups like **CAA (US) and EGAIR(EU)**, governments, and companies to protect IP and advocate for artists' style copyrights._
> - **Third,** **our work continues to focus on the perspective of human creators, aiming to ensure the rights of original authors in the era of the generative AI explosion.** We aim to establish a healthy cycle for the development of art, where creators have the right to authorize and trace their creative styles and content.
>
> **Q4: The writing of this paper has some problems.**
>
> **R4:** Thank you for pointing out them. We will improve them to make the expression clearer.
>
> [1] AI art lawsuits: Stability AI, DeviantArt, and Midjourney face litigation.
>
> [2] Inc. Getty Images (US). v. stability ai, inc. Assigned To: Jennifer L. Hall.
>
> [3] Copyright Protection: Exploring Originality and Ownership in a Digital Landscape.
>
> [4] This artist is dominating AI-generated art.
>
> [5] Glaze: Protecting Artists from Style Mimicry by Text-to-Image Models, USENIX Security 2023.

---

> > ### Comment · Reviewer_d18z · 2024-08-08
> > **My feedback**
> >
> > Thanks for the rebuttal. I have some concerns when reading other people's comments. Reviewer z2Ei mentions this proposed framework has many complex modules. By checking this paper again, I also agree with this point. If this work is for those artists, such a complex may hinder its usage. If people are reluctant to use it, any effective frameworks may become useless. How can you handle this issue? Besides, based on the provided descriptions, the efficiency of this proposed framework should also be analyzed, if the target users are those artist mentioned by the authors.

---

> ### Author Response · Authors · 2024-08-08
> **Thank you and further explanations to reviewer's feedback**
>
> Dear Reviewer **d18z**, please allow us to thank you again for reviewing our paper and for your valuable feedback. We understand your concerns and are providing additional explanations to address them.
>
> ---
>
> First, our method is both simple and efficient for users: **they only need to register the _the identifier $z$, the watermark, and the protected data with a third-party regulatory body, as mentioned in  Section 3.1 (Defender Capability) of our paper_** . [1] notes that with the further commercialization of AIGC, the standardized data flow should involve _data owners, model providers, and public regulatory agencies (trusted third parties)_. The third party will jointly hold the unique identifier $z$ and watermark with the user, ensuring a one-to-one correspondence and non-redundancy between $z$, the watermark, and the protected data. Data copyright tracing and ownership should be initiated by the user through litigation, with processes involving judicial authorization and third-party verification. Together, the identifier $z$, watermark, and protected data constitute a personal copyright entity, enabling effective and secure copyright tracing in judicial proceedings.
>
> Second, **the _essential security, rigor, and accuracy_ of personal copyright verification in judicial procedures are effectively supported and demonstrated by the rigor and complexity of the algorithmic framework presented in our paper**. We have designed a comprehensive and well-rounded protection and verification mechanism with a focus on personal copyright security, and we provide extensive experimental results to validate the reliability of our method.
>
> Third, **our framework is divided into three stages: registration, computation, and inference.** _In the registration stage_, data owners register their identifier $z$ and corresponding watermark with a third-party regulatory body. _In the computation stage_, the third party uses our algorithm to perform computations and store the results after receiving the registration list. _In the inference stage_, the framework performs copyright verification on suspicious samples and models. **We analyze the efficiency of the framework as follows**: **_On one hand_**, in terms of resource consumption, during the computation stage, using a single 3090 GPU, the average computation time per user is **1** minute, with proxy sample computations averaging **3-5** minutes and memory usage approximately **3**MB. In the inference stage, the average inference time for **1000** users ranges from **30 to 100 milliseconds** (i.e., **0.065** milliseconds per user). **_On the other hand_**, regarding copyright tracing accuracy, the ASR metric is close to **100%**, with an error rate controlled below **0.1%**, and the average watermark accuracy exceeds **99%**. Of 1000 suspicious AI mimic samples, about **97%** can be successfully verified and traced through judicial proceedings (as indicated by the t@k@100wd% metric mentioned in this paper).
>
> Users such as artists only need to register the identifier $z$ and the corresponding watermark with the third party. The design and complexity of the algorithmic framework ensure the security, rigor, and accuracy of copyright protection. Overall, our framework demonstrates its practicality in judicial security validation, resource consumption, and efficiency.
>
> [1] Building Intelligence Identification System via Large Language Model Watermarking: A Survey and Beyond.

---

> > ### Comment · Reviewer_d18z · 2024-08-09
> > **Thanks**
> >
> > Thanks for the quick reply. I will make the final decisions based on the discussions with other reviewers.

---

> > > ### Author Response · Authors · 2024-08-13
> > > **Thanks and A Gentle Reminder of the Final Feedback**
> > >
> > > Dear Reviewer **d18z**,
> > >
> > > Please allow us to thank you again for your valuable time and constructive comments. Your comments have been instrumental in helping us clarify the significance of our work and enhance its quality.
> > >
> > > _As the reviewer-author discussion phase is nearing its end_, we would like to know whether our explanations and experiments have properly addressed your concerns. We are more than happy to answer any additional questions. Your feedback will be greatly appreciated.
> > >
> > > Thank you again for your thorough review and for helping us improve our work!

---

> > > > ### Comment · Reviewer_d18z · 2024-08-13
> > > >
> > > > I do not have further questions. As we are also busy with our own submitted papers, the time is not enough for me to make my final decisions at this moment. Thanks. I will make my final decisions at the next stage.

---

### Official Review · Reviewer_jJoo · 2024-07-12

**Soundness:** 4
**Presentation:** 4
**Contribution:** 4
**Rating:** 8
**Confidence:** 5

**Summary:**

This paper introduces an innovative implicit $z$-watermarking scheme using disentangled style domains to protect dataset copyrights in text-to-image models. It achieves structured delineation of copyright boundaries, self-generalization, mutual exclusivity, and effective verification for hybrid or partial infringements. The method demonstrates high robustness and reliability against various challenges, marking a significant advancement in protecting copyrighted content in AI-generated visual data.

**Strengths:**

1. This paper designs a novel implicit $z$-watermarking scheme via disentangled style and content domains to protect dataset copyrights. Meanwhile,  instead of embedding invisible information into images, the proposed self-generalization module and mutual exclusivity module are used to explore the style boundaries.

2. This paper is very detailed and well-formulated, with precise wording, clear definitions, and easy to understand.

3. Extensive experiments demonstrate the SOTA performance of the proposed method, such as DCT-DWT-SVD, RivaGan, and SSL. Obviously, the proposed $z$-watermarking is hard to be removed by some illegal mimic models.

**Weaknesses:**

1. In table I, it is suggested to compare with some recent and SOTA watermarking methods, such as Trustmark and RoSteALS.

[1] TrustMark: Universal Watermarking for Arbitrary Resolution Images.

[2] Rosteals: Robust steganography using autoencoder latent space. In CVPRW 2023.

2. Can $z$-watermarking resist some watermark removal or attack methods, such as DDIM inversion or VAE. The authors could provide some results to improve the completeness of the experiment.

**Questions:**

Please refer to the weakness.

**Limitations:**

The author has clearly presented its limitations.

---

> ### Author Rebuttal · Authors · 2024-08-06
>
> Dear Reviewer **jJoo**, thank you very much for your careful review of our paper and thoughtful comments. We hope the following responses can help clarify potential misunderstandings and alleviate your concerns.
>
> ---
>
> **Q1:** In Table I, it is suggested to compare with some recent and SOTA watermarking methods, such as Trustmark and RoSteALS.
>
> **R1:** Thank you for your constructive suggestions! **We have added Trustmark[1] and RoSteALS[2]** as comparison baselines. Here are more details and discussions. We set up 100 images with different watermarks. Trustmark has a length of 64 bits, RoSteALS has a length of 56 bits, and ours is 128 bits. We evaluated whether the generated images contained watermarks using TP (True Positive) and TN (True Negative). For watermark extraction, we used Avg acc (%) and k@t@100%wd as evaluation metrics. The experimental results indicate that previous watermarking methods are diluted or erased easily during the generation process, which is detrimental to the traceability and ownership of the samples.
>
> | Method           | TP  | TN  | Avg acc (%) | k@t@100%wd(%) |
> |------------------|-----|-----|-------------|-------------------|
> | Trustmark | 93  | 907 | 55.37       | 6.6   |
> | RoSteALS  | -   | -   | 66.50       | 7.9 |
> | **Ours** | 1000| 0   | **99.83**   | **97.7**  |
>
> [1] TrustMark: Universal Watermarking for Arbitrary Resolution Images.
>
> [2] RoSteALS: Robust steganography using autoencoder latent space. In CVPRW 2023.
>
> ---
>
> **Q2:** Can $z$-watermarking resist some watermark removal or attack methods, such as DDIM inversion or VAE? The authors could provide some results to improve the completeness of the experiment.
>
> **R2:** Thank you for your constructive suggestions! We agree that understanding the impact of watermark removal is also important. In our paper's robustness experiments, we conducted experiments on Latent Attacks. To further demonstrate the superiority of our approach, we have included the following additional experiments. We hereby provide more details.
>
> - First, **we use the watermark removal method [1]** to attack baseline watermarking schemes and ours. For attacks using variational autoencoders, we evaluate the pre-trained image compression models: Cheng2020 [2]. The compression factors are set to 3. For diffusion model attacks, we use stable diffusion 2.0 [3]. The number of noise steps is set to 60.
> - Second, we chose Avg acc (average watermark accuracy), Detect Acc (percentage of images where decoded bits exceed the detection threshold 0.65), and k@t@100%wd as the evaluation metrics for watermark robustness. The result is as follows.
> - Third, our method achieves an average accuracy of 97.93% and 95.81%, with a detection accuracy of 100% and k@t@100%wd of 91.5% and 87.2% under VAE and Diffusion attacks, respectively. In contrast, other methods like DCT-DWT-SVD, RivaGan, and SSL show significantly lower performance. From the results, our performance significantly surpasses other watermarking schemes after being subjected to watermark removal attacks [1].
>
> | Method | Removal Attack Instance | Avg acc (%) | Detect Acc (%) | k@t@100%wd (%) |
> |-----------|-------------------------|-------------|----------------|-------------------|
> |**DCT-DWT-SVD**|VAE attack|50.17| 2.0|0.0|
> |**DCT-DWT-SVD**|Diffusion attack| 54.41 | 2.8| 0.0|
> |**RivaGan**|VAE attack|60.71|6.2| 0.0|
> |**RivaGan**|Diffusion attack|58.23|1.8|0.0|
> |**SSL**|VAE attack|62.92|15.6|0.0|
> | **SSL**|Diffusion attack|63.21|16.3|0.0|
> |**Ours**|VAE attack|**97.93**|**100** |**91.5**|
> |**Ours**| Diffusion attack| 95.81|100|87.2|
>
> [1] Zhao X, Zhang K, Su Z, et al. Invisible image watermarks are provably removable using generative ai[J]. arXiv preprint arXiv:2306.01953, 2023.
>
> [2] Z. Cheng, H. Sun, M. Takeuchi, and J. Katto, “Learned image compression with discretized gaussian mixture likelihoods and attention modules,” in Proceedings of the IEEE/CVF Conference on Computer Vision and Pattern Recognition, 2020, pp. 7939–7948.
>
> [3] R. Rombach, A. Blattmann, D. Lorenz, P. Esser, and B. Ommer, “High-resolution image synthesis with latent diffusion models,” in Proceedings of the IEEE/CVF Conference on Computer Vision and Pattern Recognition, 2022, pp. 10 684–10 695.

---

> > ### Comment · Reviewer_jJoo · 2024-08-10
> >
> > Thank you for your detailed response. The additional experiments and explanations have addressed most of my concerns. I believe this paper presents an interesting and practically valuable work, with performance surpassing similar digital watermarking methods, making it applicable for copyright protection in generative AI. Considering the superior performance and the novelty of this approach, I decide to raise my score.

---

### Official Review · Reviewer_z2Ei · 2024-07-12

**Soundness:** 3
**Presentation:** 2
**Contribution:** 3
**Rating:** 4
**Confidence:** 3

**Summary:**

While text-to-image models excel in generating high-quality images, they also raise issues of unauthorized dataset copyright protection. This paper proposes a novel implicit watermarking scheme that detects and protects dataset copyrights by disentangling the style domain  to generate watermarks. The proposed method achieves One-Sample Verification, significantly improving existing copyright protection mechanisms.

**Strengths:**

1. The authors are the first to utilize the disentangled style domain to detect unauthorized dataset usage in text-to-image models, and they have effectively implemented a method called z-watermarking to enhance copyright protection.

2. The authors conducted comparative experiments with state-of-the-art protection methods and robust experiments under various conditions. They also performed comprehensive ablation studies, demonstrating the effectiveness and robustness of their method. The ablation studies verify the importance and effectiveness of each module proposed by the authors.

**Weaknesses:**

1. This paper involves many complex modules, and in Section 3 (Method), the authors use many symbols and subscripts. However, the annotations for these symbols and subscripts are not very clear. For instance, in line 144, $\mathcal{E}_z(z_x|(x, ϕ), z) = s$. It is not clear what the input to the style domain encoder is. If ‘|’ denotes a probabilistic condition, then the style domain encoder only has one input, but Figure 2 appears to show two inputs, which is confusing. For the numerous symbols and labels, it is recommended that the authors provide a unified introduction in each section.

2. The paper introduces the operational flow of the model in three parts. However, the authors seem to focus more on explaining each module rather than the connections between modules and the overall operational flow. As a result, after reading these parts, it remains difficult to grasp the overall process of the proposed method. The paper appears technically sound, but there is still significant room for improvement in introducing the technical flow.

**Questions:**

1. In line 150, “Moreover, we implement layer-wise guidance dropout by selectively zeroing out portions of  s_{1:m} , thereby diminishing the decoder’s dependency on sub-vector correlations.” Could you clarify the criteria used for this selection? Additionally, how does this approach effectively reduce the decoder’s dependency on sub-vector correlations? Is there existing literature that supports this conclusion, or can you provide a more detailed explanation?

2. In the experimental section, line 216, the authors tested “17 artists (e.g., Van Gogh and Monet) and 10 AI artworks.” Given that copyright protection is an incremental task, as new artists or AI artworks are created, the number of protected entities will increase. This presents two potential challenges:
	1. How does the proposed method handle the increasing number of protected entities? Does it require retraining with each addition, or is there a more cost-effective solution?
	2. As the number of protected entities grows, will the styles of these entities influence each other, potentially reducing the model’s detection performance?

**Limitations:**

The authors have addressed the limitations.

---

> ### Author Rebuttal · Authors · 2024-08-06
>
> Dear Reviewer **z2Ei**, thank you very much for your careful review of our paper and your thoughtful comments. We hope that the following responses will help clarify any potential misunderstandings and alleviate your concerns.
>
> ---
>
> **Q1:** Regarding “Moreover, we implement layer-wise guidance dropout by selectively zeroing out portions of $s_{1:m}$, thereby diminishing the decoder’s dependency on sub-vector correlations.” Is there existing literature that supports this conclusion? Could you provide a more detailed explanation?
>
> **R1:** Thank you for your comments and we do understand your concerns. To further alleviate your concerns, we provide more explanations.
> - Firstly, **our goal is to achieve a bidirectional mapping between images and disentangled variables $s_{1:m}$.** By reducing the co-adaptations between Unet layers, it enhances the model's generalization ability, meaning that neurons are less likely to rely too much on each other, thereby achieving decoupling. The dropout guided by zeroing out disentangled variables during training essentially aims to encourage the model to obtain linearly independent solutions for $s_{1:m}$.
>
> - Second, **the latest paper SODA[1]** (presented at _CVPR 2024_, a self-supervised diffusion model designed for representation learning) suggests that disentangled latent spaces can better represent the generated images. **In the conclusion of the reference[1]: To improve localization and reduce correlations among the sub-vectors, we present layer masking – a layer-wise generalization of classifier-free guidance [2].** The reference provides ample ablation experiments to validate it.
>
> - Third, we have also thoroughly validated the correctness of this conclusion in our experiments. The results presented in Table 1, Table 2, and Figure 3 of our main experiments in the paper demonstrate the rationale behind this conclusion.
>
> [1] Hudson D A, Zoran D, Malinowski M, et al. Soda: Bottleneck diffusion models for representation learning[C]//Proceedings of the IEEE/CVF Conference on Computer Vision and Pattern Recognition. 2024: 23115-23127.
>
> [2] Ho J, Salimans T. Classifier-free diffusion guidance[J]. arXiv preprint arXiv:2207.12598, 2022.
>
> ---
>
> **Q2:** How does the proposed method handle the increasing number of protected entities? Does it require retraining with each addition, or is there a more cost-effective solution?
>
> **R2:** We use the identifier $z$  with effectively infinite capacity to handle the increasing number of protected entities. This approach does not require retraining the style domain encoder and incurs only minimal additional cost. We will provide more explanations.
>
> - **First, one of the advantages of the paper is to use the _identifier $z$_ to address the issue of the increasing number of protected entities.** In this paper, unrestricted $z$ can represent any text, image, video, or audio, which is encoded into $z_{\text{emb}}$ and injected into the style domain to ensure the boundary of the protected unit.
>
> - **Second, $z$ signifies the identifier that maximally shifts the contraction domain to the edge distribution** of the style representation space. After decoupling the style domain and negative samples and performing dynamic contrastive learning to increase the distance in the similarity space, $z$ is further shifted to the boundary space by injecting $z$.
>
> - **Third, we do not need to retrain the style domain encoder.** We only need to decouple the protected unit into the style domain, inject the corresponding identifier $z$ to shift it to the edge distribution, and store the mapping relationship in the watermark extractor (This only incurs a minimal cost).
>
> ---
>
> **Q3:** As the number of protected entities grows, will the styles of these entities influence each other, potentially reducing the model’s detection performance?
>
> **R3:** Thank you for your comments and we do understand your concerns. In response to Question 2, **the advantages of our paper for addressing this issue are to decouple the style domain, utilize dynamic contrastive learning, and use the unique and  infinite identifier $z$.** To further alleviate your concerns, we provide more explanations.
> - First, we decouple the latent variables into sub-vectors that control image generation, thereby extracting linearly independent combinations of the image's essential features. Then, we utilize dynamic contrastive learning to set **_sample anchors, edge samples, central samples, and negative samples_** for the protected units, encouraging the style domain of the protected units to occupy mutually exclusive regions in the high-dimensional space.
>
> - Second, we propose the identifier $z$ to further enhance the reliability and security of the solution. $z$ represents **_the unique and critical identifier_** that maximally shifts the contraction domain to the edge distribution of the style representation space. Since $z$ is an arbitrary identifier (including any text, string, image, etc.), its capacity is effectively infinite, which is sufficient to differentiate the growing number of protected entities.
>
> - Third, we have thoroughly validated the feasibility of our approach through main and ablation experiments. **As the number of protected entities increases, the distinctiveness of the model remains robust, ensuring that the styles of these entities do not influence each protected unit.**
>
> |128bit| 0-20% | 20-40%|40-60%|60-80% | 80-90%|90-100% | TP|TN| Avg acc (%) | k@t@100%wd (%)  |
> |-----------|-------|--------|--------|--------|--------|---------|-----|-----|---------------|------------------|
> |**$z$ Error**| 0 |151|555|291| 3| 0 | 0 |1000|52.15|0|
> |**$z$-watermarking**|0| 0 |0 |1| 6| **993** | 1000|0| **99.87** | **97.9** |
>
> ---
>
> **Q4:** The annotations for these symbols and subscripts are not clear, and there is still significant room for improvement in introducing the technical flow.
>
> **R4:** Thank you for pointing out the shortcomings in our writing. We will make revisions.

---

> ### Author Response · Authors · 2024-08-13
>
> Dear Reviewer **z2Ei**,
>
> Thank you once again for your valuable time and constructive comments. We would like to kindly inform you that we have already addressed your concerns in our rebuttal.
>
> As the reviewer-author discussion phase is nearing its end, we would like to know whether our explanations and experiments have properly addressed your concerns. We are more than happy to answer any additional questions. Your feedback will be greatly appreciated.

---

### Author Rebuttal · Authors · 2024-08-06

## Global Response

We would like to express our gratitude to all reviewers for their thorough reading and constructive feedback. Since all reviewers have expressed several main concerns, we will try to address these issues in the global response.

**Rethinking Personal Data Copyright Ownership in the Era of Generative AI.** Due to the explosive growth of generative AI, an increasing number of creators' works, including creative entities, brushstrokes, and styles, are being used for unauthorized profit. In Glaze's survey, based on responses from 1,207 artists, the vast majority hope for fair legislation to protect the unique artistic styles and content of their works. However, unfortunately, there are currently no feasible solutions, and this issue is highly challenging. Currently, generative AI is being maliciously used by some to easily learn, imitate, and plagiarize unauthorized human works for profit. This severely undermines creators' motivation, damages their creative enthusiasm, and turns high-quality works into others' benefits. This step may be urgent for ensuring intellectual property rights for human creativity in the AI era. Therefore, we aim to establish a positive and healthy cycle for the development of art between generative AI and human creation.

**$z$ identifier causes exclusivity.** In this paper, $z$ is designed to ensure the exclusivity and uniqueness of the protected unit distribution within large datasets. $z$ signifies the identifier that maximally shifts the contraction domain to the edge distribution of the style representation space, after decoupling the style domain and performing dynamic contrastive learning to increase the distance in the similarity space. Since $z$ is an arbitrary identifier (such as text, strings, images, etc.), its capacity is effectively infinite, which further enhances the reliability and security of the solution. In machine learning, there are inherent differences in the high-dimensional feature distributions of protected units. Our approach, which utilizes the $z$ identifier, decouples the style domain and employs dynamic contrastive learning, aims to shift this distribution. In the paper, we provide hypothesis testing (Section 4) and experimental data (Section 5.3) to demonstrate the feasibility of our proposed solution.

|                    | 0-20% | 20-40% | 40-60% | 60-80% | 80-90% | 90-100% | TP | TN | Avg acc (%) | k@t@100\%wd (%)  |
|--------------------|-----------|------------|------------|------------|------------|-------------|--------|--------|----------------------------------|-----------------------------------|
| $z$ Error          | 0         | 151        | 555        | 291        | 3          | 0           | 0      | 1000   | 52.15                            | 0                                 |
| $z$-watermarking   | 0         | 0          | 0          | 1          | 6          | **993**     | 1000    | 0      | **99.87**                        | **97.9**                     |

**Regarding the baselines for watermarking and backdoor methods.** We compared digital watermarking methods such as DCT-DWT-SVD, RivaGan, SSL, Trustmark, and RoSteALS, as well as the backdoor-based method DIAGNOSIS. The experimental results indicate that previous watermarking methods are diluted or erased during the diffusion generation process, which is detrimental to the traceability and ownership of the sample. Additionally, we introduced watermark removal attacks (such as VAE and Diffusion Attack), where digital watermarking methods nearly fail, while our approach still demonstrates strong robustness.

---

### Comparison of Methods

| Method         | TP  | TN  | ASR | Avg acc (%) | $k@t@100\%wd$ (%) |
|----------------|-----|-----|-----|-------------|-------------------|
| DCT-DWT-SVD    | -   | -   | -   | 57.76       | 0.1               |
| RivaGan        | -   | -   | -   | 61.34       | 0.1               |
| SSL            | -   | -   | -   | 64.39       | 0.1               |
| Trustmark      | 93  | 907 | 9.3 | 55.37       | 6.6               |
| RoSteALS       | -   | -   | -   | 66.50       | 7.9               |
| DIAGNOSIS      | 993 | 7   | 99.3| -           | -                 |
| **Ours**       | **1000** | 0 | **100** | **99.83** | **97.7**         |

---

### Removal Attack Performance

| Method         | Removal Attack Instance | Avg acc (%) | Detect Acc (%) | k@t@100 %wd (%) |
|----------------|-------------------------|-------------|----------------|-------------------|
| DCT-DWT-SVD    | VAE attack              | 50.17       | 2.0            | 0.0               |
|                | Diffusion attack        | 54.41       | 2.8            | 0.0               |
| RivaGan        | VAE attack              | 60.71       | 6.2            | 0.0               |
|                | Diffusion attack        | 58.23       | 1.8            | 0.0               |
| SSL            | VAE attack              | 62.92       | 15.6           | 0.0               |
|                | Diffusion attack        | 63.21       | 16.3           | 0.0               |
| **Ours**       | VAE attack              | **97.93**   | **100**        | **91.5**         |
|                | Diffusion attack        | **95.81**   | **100**        | **87.2**         |

---

### Decision · Program_Chairs · 2024-09-25

**Decision:**

Accept (poster)

**Comment:**

1x A and 3x BR. This paper proposes an implicit watermarking scheme using disentangled style domains to protect dataset copyrights in text-to-image models. The reviewers agree on the (1) important topic, (2) novel method, and (3) comprehensive experiments. Most of the concerns, such as the efficiency analysis, have been addressed by the rebuttal. Therefore, the AC leans to accept this submission.